# A review of current and possible future human-water dynamics in Myanmar's river basins

Linda Taft[1*], Mariele Evers[1]

[1]Department of Geography, University of Bonn, Meckenheimer Allee 166, 53315 Bonn, Germany

*Corresponding author: L. Taft (ltaft@uni-bonn.de)

**Abstract**

Rivers provide a large number of ecosystem services and riparian people depend directly and indirectly on water availability, quality and quantity of the river waters. The country's economy, the people's well-being and income particularly in agriculturally dominated countries is strongly determined by the availability of sufficient water. This is particularly true for the country of Myanmar in Southeast Asia, where more than 65% of the population live in rural areas, working in the agricultural sector. Only few studies exist on river basins in Myanmar at all and detailed knowledge providing the base for human-water research is very limited. A deeper understanding on human-water system dynamics in the country is required because Myanmar's society, economy, ecosystems and water resources are facing major challenges due to political and economic reforms and massive and rapid investments from neighbouring countries. However, not only policy and economy modify the need for water. Climate variability and change is another essential driver within human-water systems. Myanmar's climate is influenced by the Indian Monsoon circulation which is subject to interannual and also regional variability. Particularly the central dry zone and the Ayeyarwady delta are prone to extreme events such as serious drought periods and extreme floods. On the one hand, the farmers depend on the natural fertilizer brought by regular river inundations and high groundwater level for irrigation; on the other hand, they suffer from these water-related extreme events. It is expected that theses climatic extreme events will likely increase in frequency and magnitude in the future as a result of global climate change. Different national and international interests in the abundant water resources may provide opportunities and risks at the same time for Myanmar. Several dam projects along the main courses of the rivers are currently in the planning phase. Dams will most likely modify the river flows, the sediment loads and also the still rich biodiversity in the river basins, in an unknown dimension. Probably, these natural and anthropogenic induced developments will also impact a special type of farming, we call it alluvial farming, in the river floodplains and on sandbars in the Ayeyarwady River basin in Myanmar, which is called Kaing and Kyun, respectively.

Relevant aspects for future development of Myanmar's river basins combine environment-water-
related factors, climate, economic and social development, water management and land use changes.
Research on this interplays need to capture the spatial and temporal dynamics of this drivers. Yet, it is
only possible to gain a full understanding of all these complex interrelationships, if multi-scale
spatiotemporal information will be analysed in an inter- and transdisciplinary approach. This paper
gives a structured overview on the current scientific knowledge available and reveals the relevance of
this information with regard to human-environment and particularly to human-water interactions in
Myanmar's river basins. By applying the eDPSIR framework, it identifies key indicators in the
Myanmar human-water system, which has been shown exemplary by giving an example of use related
to alluvial farming in the central dry zone.
**Keywords**
Human-water dynamics; River basins; Myanmar; Southeast Asia, Climate Change, eDPSIR, Alluvial
farming
# 1 Introduction
Rivers provide a large number of ecosystem services, e.g. water supplies, food source, biodiversity
conservation or drought mitigation, and river basins are home to almost one billion people worldwide
(Postel and Richter, 2003; Allen et al., 2010; Di Baldassarre et al., 2013). Riparian people depend
directly and indirectly on water availability, quality and quantity of the river waters, which is, in turn,
influenced by precipitation, evaporation, glacial meltwater in the river source areas, and increasing
human impact. For example, pollution, increasing water use for irrigation and dam building alter the
quality and availability of river waters. Additionally, anthropogenic climate change will possibly
impact flow regimes and the water demand. Rivers worldwide are under pressure due to multiple uses
which often have severe impacts on ecosystems, or water quality and flow. Vörösmarty et al. (2010)
stated that 65% of global river discharge, and the aquatic habitats supported by this water, are under
moderate to high threat of biodiversity loss.
The country's economy, the people's well-being and income particularly in agriculturally dominated
countries strongly depend on the availability of sufficient water. This is particularly true for the
country of Myanmar in Southeast Asia, where more than 65% of the population live in rural areas,
working in the agricultural sector (FAO, 2014). The Ayeyarwady River (also referred to as Irrawaddy
River) catchment covers 413,700 km² which represents about 61% of the country. Thus it is the most
important river system in Myanmar. The mighty Ayeyarwady River is called the life line of the nation
because it serves for transportation, domestic and industrial water supply, irrigation, a high
biodiversity and fishing. This river is highly important for Myanmar but is the least known among the

large Asian rivers (Furuichi et al., 2009). Since the end of Myanmar's political and economic isolation in 2011, the country's abundant water resources are now facing major changes. It is assumed that the current and future progressive socio-economic development of the country have and will have a significant impact on the water resources (Kattelus et al., 2014). However, not only political, economic and demographic changes have and will have major effects on the natural water resources. The headwaters of the Ayeyarwady River are fed by glacier melt in the Himalayan Mountains and the river discharge is likely to change due to climate change impacts. Myanmar's climate is directly influenced by the Indian summer monsoon (ISM; Sen Roy and Kaur, 2000; Sein et al., 2015) which is the second basic source of Myanmar's rivers. It is currently still not predictable if the complex Asian monsoon circulation will strengthen, weaken or become more variable as a result of global warming (Turner and Annamalai, 2012; IPCC, 2014). Already now, there seems to be a trend to a delay and an earlier ending of the monsoon rains of 2 weeks in Myanmar respectively (The Irrawaddy, 2015). On the one hand, the Burmese riparian people depend on frequent river floods for agriculture, particularly rice production in the river delta regions. On the other hand, extreme flood events can cause destructing effects. Just recently, the western part of the country was affected by very heavy monsoon rains in August 2015. Thousands of people had to be evacuated and more than 100 people died (Burki, 2015). The occurrence of extreme weather events like floods, cyclones and severe droughts has shown an increasing trend over the last six decades in Myanmar, most likely as a result of climate change (GCCA, 2012).

Only few research exist on river basins in Myanmar at all and detailed knowledge is very limited (e.g. Varis et al., 2012; Salmivaara et al., 2013). However, more research on human-water dynamics in the country is strongly required because Myanmar's society, economy, ecosystems and water resources are facing major challenges due to political and economic reforms, massive and rapid investments from neighbouring countries (particularly from China and Thailand; Webb et al., 2012) and climate change impacts. There is a large number of grey literature such as reports and workshop presentations from NGOs, political institutions and Burmese and international organizations, dealing with human and climate impacts and the water resources in Myanmar. However, systematically compiled English scientific publications on this topic have not been published yet. Hence, English grey literature and peer-review publications on current and likely future impacts from human activities and climate change on Myanmar's river basins have been reviewed by the authors in order to gain an overview on the key drivers in these human-water dynamics.

Following a socio-ecological understanding we hypothesise that all components (e.g. stakeholders, water resource, climate, aquatic fauna) within the Burmese river basins interact and that the degree of interactions between driving forces (e.g. foreign investments, water demand, water management measures, regional climate change) and feedbacks permanently change. If, for example, the demand

for domestic water increases due to a dry spell, the riparian people will increasingly extract river water
as well as ground water. This higher water extraction in turn, will probably impact the aquatic
ecosystem which has potential negative effects on the fisheries and therefore also on the economic
income of the people.
The major aim of this review was to compile the natural given physical conditions and the socio-
economical features in terms of land and water use in order to make this information internationally
accessible in a scientific status-quo review paper. The first part of the paper provides general
information on physical features with focus on the river basins, followed by socio-economical
features. Chapter 4 concentrates on possible future impacts with focus on climate change. Based on
the reviewed literature we attempt to structure the information on human-water dynamics by means of
the eDPSIR (enhanced driving force-pressure-state-impact-response) framework by Niemeijer & de
Groot (2008a, b). We applied this framework by using a concrete example in order identify important
key nodes within a causal network of various driving forces, feedback mechanisms, impacts and
responses on the societal and the physical-environmental side. Open research gaps concerning human-
water dynamics in Myanmar and recommendations for research approaches have been identified and
elaborated in the end.
**2    Physical features**
The Republic of the Union of Myanmar (9°55' - 28°15' N,  92°10' - 101°11' E) is a Southeast Asian
country located between Bangladesh and India to the west, China to the north and northeast, Laos and
Thailand to the east and the Bay of Bengal and the Andaman Sea to the southwest and south (Fig. 1).
The maximum north-south extent is about 2,500 km and the maximum west-east extent is ca. 900 km.
With 676.578 km$^2$ Myanmar (Department of Population, 2015) is the second largest country in
Southeast Asia after Indonesia.
2.1    Geology and geomorphology
The country slopes downward in elevation from north to south and the central lowlands are
surrounded by steep mountain ranges (Fig.1). Three mountain ranges trending from north to south,
namely the Rakhine Yoma (the term Yoma means mountain range), the Bago Yoma and the Shan
Plateau (from west to east), divide the country (Fig. 1). The Rakhine and the Bago mountain ranges
have been thrusted up through the collision of the Indian-Australian and the Eurasian plate since the
past 50 million years (Bender, 1983). The Shan Plateau was already formed during the Mesozoic era
and it has an average elevation of about 900 m a.s.l. (Hadden, 2008). The topography can be divided
into five sub-regions: 1) the northern mountains including the highest point of Myanmar Mount
Hkakabo Razi (5,881 m a.s.l.); 2) the western Rakhine ranges; 3) the eastern Shan Plateau; 4) the

central basins and lowlands and 5) the coastal plains including the wide Ayeyarwady delta. The Mount Hkakabo Razi is part of a geological complex where the Indian-Australian plate has been colliding with the southern edge of the Eurasian plate since the Eocene (Hadden, 2008). This northern mountain region is the source area of several of Asian's great rivers, including the Irrawaddy. The central basin lies between the western Rakhine ranges and the eastern Shan Plateau in the rain shadow of the monsoon precipitations. The Ayeyarwaddy, Chindwin and Sittaung rivers cover soft sandstones, shales and clays with their fertile alluvial deposits in the central basin (Bender, 1983). The coastline has a length of about 3,000 km and there are numerous islands of varying sizes (Oo, 2002).

## 2.2 Hydrogeography

Major rivers are the Ayeyarwady, the Salween, the Chindwin and the Sittaung. All these rivers are understudied river basins (Salmivaara et al., 2013), despite their great importance for the Burmese people's life and the nation's economy. The north-south trending courses of most of the Burmese rivers are geologically predetermined following the mountain ranges Rakhine, Bago and the Shan Plateau. For about 230 km, the transnational Mekong River forms the border between Myanmar and Laos (Fig. 1).

The Ayeyarwady River is Myanmar's most important commercial waterway (Salween Watch and SEARIN, 2004). It is about 2,170 km long and originates at the confluence of the Mali Hka and N'Mai Hka rivers in the northern Kachin state (Fig. 1). The headwaters of both rivers originate in the eastern syntaxis of the Himalayas and the Tibetan Plateau in Yunnan Province, China. The river basin of the Ayeyarwady covers around 413,700 km² of which 95% is located in Myanmar (Salween Watch and SEARIN, 2004; Bird et al., 2008). The broad fertile lowland floodplain is extensively used for agriculture. The river is fed by glacial meltwater in the source areas of the Mali Hka and N'Mai Hka rivers as well as by precipitation. Based on data collection between 1969-1996 by Furuichi et al. (2009), the average annual discharge is $379 \pm 47$ x $10^9$ m³/year and around 70% of it occurs between July and October (Robinson et al., 2007). The Ayeyarwady has the fifth highest sediment load of any major river worldwide (Furuichi et al., 2009; The World Bank, 2014). Furuichi et al. (2009) estimated the suspended sediment load to be $325 \pm 57$ x $10^6$ t annually. However, the river is navigable year-round for approximately 1,500 km from Yangon, but sandbanks and shallow sections make it often difficult to navigate during the dry season (Lwin, 2014). The basin's ecosystem is very rich and dynamic and the river is home to the endangered 'Irrawaddy dolphin' (Smith et al., 2009; Aung et al., 2013).

With a total length of about 2,800 km the transboundary (China, Thailand, Myanmar) Salween River is one of the longest rivers in Southeast Asia. However, it is navigable for only 150 km from its delta due to its rapids and deep gorges (Salween Watch and SEARIN, 2004). Annual runoff is

approximately 210 km³ (Robinson et al., 2007). The source of the river is located on the Tibetan
Plateau and subsequently the water flows through Yunnan Province in China to the eastern part of
Myanmar where the Salween drains the Shan Plateau. For approximately 120 km, the river forms the
border between Myanmar and Thailand until it flows to the Andaman Sea in the Gulf of Martaban
(Salween Watch and SEARIN, 2004; Fig. 1). The river basin covers 320,000 km² and has one of the
most diverse ethnic concentrations worldwide (Salween Watch and SEARIN, 2014). Furthermore, the
basin is very rich in natural resources including surface and groundwater, forest, wildlife, fisheries and
minerals (FAO, 2011).
The Chindwin River exists since at least the Eocene and is the largest tributary of the Ayeyarwady
(Hedley et al., 2010). It has a length of about 1,200 km (Salween Watch and SEARIN, 2014). The
Chindwin rises in the Kumon Range in northern Myanmar and reaches the Ayeyarwady near
Mandalay in the central dry zone. For about 600 km the river is navigable from its confluence with the
Ayeyarwady River (Ministry of Forestry, 2005). Most of its course has not been studied yet due to the
difficulty of access (Salween Watch and SEARIN, 2004).
The Sittaung River originates at the southern edge of the Shan Plateau and drains after 420 km into the
Gulf of Martaban of the Andaman Sea (Salween Watch and SEARIN, 2004). Year-round, the Sittaung
River is navigable only for 40 km and for 90km during the rainy season. It is mainly used for floating
teak wood for export to the souths. At its lower course, the river is linked by a canal to the Bago River,
located in Yangon (Fig. 1).
The Ayeyarwady delta is one of the major tropical deltas worldwide (Hedley et al., 2010). Its current
extensive wedge-shape originated around 7.000-8.000 years ago and it comprises >20,500 km² of flat,
low-lying fertile delta plain with five major tributaries (Hedley et al., 2010; Woodroffe, 2000). The
delta area continues upriver at sea-level for more than 200 km (Webster, 2008). The delta plain hosts a
fragile and complex ecosystem of mangrove swamps and tidal estuaries (Salween Watch and
SEARIN, 2004). Mangrove forests play an important role in delta evolutions because they act as
sediment traps, primary colonisers and bio shields against impacts of cyclones and tsunamis. However,
the ecological status of the Ayeyarwady mangroves is continuously declining due to increasing rice
production, land use changes and population growth (Ministry of Forestry, 2005; Webb et al., 2014).
The Ayeyarwady delta is under intensive land use and the population density is the second highest
(177/km²) in the entire country, after Yangon (716/km²) (Salmivaara et al., 2013; Department of
Population, 2015). Saline water penetrates up to 100 km upstream due to tidal influences (Aung, 2003
in Hedley et al., 2010). Drainage, flood protection and salt intrusion are major concerns in the
Ayeyarwady delta (FAO, 2015). The Salween River has rather a river mouth than a clearly developed
delta and is less populated. However, the Salween river mouth area is facing similar environmental
pressures, only on a smaller scale (Salmivaara et al., 2013).
2.3   Soil types
Due to the wide range of climatic and geologic conditions, soil types in Myanmar vary accordingly.
Fertile alluvial soils are predominantly located in the river basins of the Ayeyarwady, the lower
Salween and the Chindwin Rivers (MOAI, 2001).  These soils are of high importance for farming (see
3.1). Red-brown and yellow-brown forest soils (cambisols following the FAO soil classification or
inceptisols following the USDA soil classification) are found in the hilly areas of the mountains ranges
and its forelands. These soils are suitable for forest plantation (Ministry of Forestry, 2005). The central
part of the country is covered with red-brown and dark compact savanna soils which are susceptible
for soil erosion and dryland salinity. The humus content of red earths is relatively high ($< 8\%$) and
thus this soil type is very suitable for diversified agriculture which can be found from the eastern
Mandalay division to large parts of the Shan Plateau (Ministry of Forestry, 2005).
2.4   Climate
Few regional studies exist on modern climate conditions in Myanmar. In general, large parts of the
country have a tropical monsoon climate. Due to the diverse orography of the country ranging from
low-lying delta regions to high mountainous terrain, the climate can be divided into the following five
sub-types according to the Köppen-Geiger climate classification (Peel et al., 2007): 1) Tropical,
monsoon climate (Am) along the coastlines and the western part; 2) Tropical, savannah climate (Aw)
in the central and eastern part; 3) Temperate, dry winter, hot summer climate (Cwa) in the north-
eastern mountainous area; 4) Temperate, dry winter, warm summer climate (Cwb) in the northern part,
a small area subsequent to the Cwa climate region and 5) Temperate, without dry season, warm
summer (Cfb) in the most north-eastern high mountain area.
Palaeoclimate research in Myanmar is very scarce, although findings about past monsoon variabilities
in this region would definitely contribute to a deeper understanding of this atmospheric circulation.
There are teak tree ring chronologies covering the last three centuries in Myanmar (D'Arrigo et al.,
2011; D'Arrigo and Ummenhofer, 2015). Following these studies, the tree-ring records show monsoon
rainfall variabilities consistent with results from surrounding countries, indicating that Myanmar is
influenced by the same atmospheric circulation system. Sen Roy and Kaur (2000) noted that even
though India and Myanmar are geographical neighbours and are influenced by the same monsoon
system, Myanmar's rainfall seems to have no significant relationship with the rainfall of Northern
India. This pattern might be due to the fact that the Rakhine Mountains ($< 3,800$ m a.s.l.) located in the
western part of Myanmar (Fig. 1) redirect the wind flows. In contrast, D'Arrigo et al. (2011) detected
a positive correlation of monsoon variability in Myanmar with the monsoon larger scale indices over
northeastern India based on teak tree ring chronology for the last three centuries. These contrary
findings highlight the urgent need for more climate research in Myanmar.
2.4.1    Precipitation
Myanmar's climate is largely influenced by the Indian summer monsoon as well as from convective
rainfall from the Bay of Bengal (Sen Roy and Kaur, 2000; D'Arrigo et al., 2011; Htway and
Matsumoto, 2011; Sein et al., 2015). The patterns of rainfall indicate considerable complexity,
particularly in summer, when Indian and East Asian monsoon circulations interact (D'Arrigo et al.,
2011). Already Maung (1945) studied the forecasting of coastal monsoon rainfalls in Myanmar;
however, his study does not include a detailed description of the general climatology. Sen Roy and
Kaur (2000) gave an overview on the climatology of monsoon rains of Myanmar using 33 years
(1947-1979) of station level monthly data. After this study, about 75% of the country's annual average
rainfall is from June to September (Sen Roy and Kaur, 2000). Sein et al. (2015) concluded that the
summer monsoon accounts for almost 90% of Myanmar's observed rainfall. The monsoon rains reach
the southern part of Myanmar by around the third week of May and cover the entire country by the
beginning of June (Sen Roy and Kaur, 2000). Results of a study by Sen Roy and Sen Roy (2011)
showed the existence of five homogenous precipitation regions, namely, north, west, central, east and
south Myanmar. Thereby, the amount of annual precipitation varies between 500-1.000 mm in the
central dry zone (Johnston et al., 2013; FAO, 2015) and up to 4.000-6.000 mm at the western coast
(MOAI, 2001; FAO, 2015). The central dry zone lies in the rain shadow of the Rakhine Mountains
located along the western coastline (Fig. 2). This area receives only 3.2% of the country's total rainfall
(Ministry of Forestry, 2005). Easterly winds and local depressions in the Gulf of Thailand can cause
post-monsoon rains from mid-October to end-November (MOAI, 2001; Sein et al., 2015).A
correlation between El Niño-Southern Oscillation (ENSO) and the variability of Asian monsoon
intensity has been discussed elaborately during the last decades (e.g. Kumar et al., 1999; Torrence and
Webster, 1999; Xavier et al., 2007; Li and Ting, 2015). All these studies conclude a significant
correlation between both atmospheric circulations. Current research from Sein et al. (2015) indicated
that El Niño events can result in drought periods in Myanmar, while La Niña events can result in more
extreme floods due to intensified monsoon rains. Temperature
The average temperature varies from 21-34°C in the hot season and from 11°C-23°C in the cool
season, depending on location and elevation. The mean relative humidity ranges between 58 and 79%
(Ministry of Forestry, 2005). Average diurnal temperatures show little variation across the country
ranging from 26°C-28°C between Sittwe in the western region, Yangon near the southern coast and
Mandalay in the central dry zone. During the rainy season, the diurnal temperatures range between 25-
33°C and from 10-25°C during the cold season. Between mid-April and mid-May, the maximum
temperatures rise continuously in the whole country (Htway and Matsumoto, 2011). The maximum
diurnal temperatures in the central dry zone can reach >43°C in the hot season prior to the monsoon
season (Aung, 2002). In this area, the mean monthly potential evapotranspiration exceeds the mean
monthly rainfall.
## 2.5 Hydro-meteorological extreme events and climate variability
Myanmar is considerable prone to risks from weather extremes and climate variability. According to
the Germanwatch Global Climate Risk Index, Myanmar is one of the countries worldwide affected
most by extreme weather events between 1993 and 2012 (Kreft and Eckstein, 2014). The coast, the
river delta zones and the central dry zone are the most vulnerable areas for weather extreme events
like cyclones, river floods, storm surges and drought periods. Climate variability is a major concern
for the country since the majority of Myanmar's economy and people's income and wellbeing are
depending on the right timing and amount of monsoon rains. Myanmar's farmers strongly depend on
monsoon precipitation since they use the water for irrigating rain-fed rice paddies and storing the rain
water for the dry season. However, extreme amounts of monsoon rains have the potential to destroy
their livelihoods. Extreme and long-lasting dry periods or extreme low amounts of monsoon rains
cause water scarcity and threaten the food security of the country.
### 2.5.1 Floods
Floods can represent both a basic asset for people's well-being, income and cultures, but also a
drawback for a societal and economic development. Myanmar is regularly affected by severe floods
comprising river floods, flash floods, pluvial floods and coastal floods. Catastrophic flash floods
associated with high rainfall occurred in the central dry zone e.g. in the year 2011(Rao et al., 2013).
Just recently, the western part of the country was affected by very heavy monsoon rains in August
2015. Particularly, the Ayeyarwady delta zone and the central dry zone are extremely vulnerable to
impacts from floods due to associated crop loss and the relatively dense population. In hilly and
mountainous rural areas, heavy rainfalls often trigger disastrous landslides with severe consequences
for the Burmese people who normally live in small wooden huts. The flood risk of Myanmar is
assessed very high due to high vulnerability and low capacity to cope with floods. For the future, the
frequency of 100-year floods in Myanmar is likely to increase (Hirabayashi et al., 2013).
### 2.5.2 Droughts
Increasing pressure on water resources and water scarcity is becoming a worldwide problem in most
arid and semi-arid regions (Kahil et al., 2015). Particularly in the central dry zone of Myanmar,
rainfall is associated with high heterogeneity across space and time (McCartney et al., 2013).
Precipitation amounts in the dry zone are generally less compared to other regions in Myanmar (see
chapter 2.2.1). In the here presented context, a drought is considered as a temporary extreme dry
period characterized by below-normal precipitation over a period of months or even years (Dai, 2011).
Severe drought periods in e.g. the years 1997-98, 2010 and 2014 led to crop failures and water
shortage in the central dry zone where more than 14 million people predominantly practice agriculture.
Most of the wells dried up due to the sinking of groundwater levels (Department of Meteorology and
Hydrology Myanmar DMH, n.d.). Due to a strong El-Niño impact since 2015, the country, and
particularly the dry zone and the Ayeyarwady delta, is severely affected by drier than average
conditions associated with risks such as fire hazards, drought, disease and food insecurity (FAO,
2016). The sources of income are affected by drought periods as well as the quality and availability of
domestic and drinking water which can have severe effects on people's health. Droughts can also have
negative impacts on the river basin's ecosystem (Kahil et al., 2015). During drought periods the
navigability of the rivers is a severe problem for national and international companies as well as for
the people living in this area (The World Bank, 2014; Ministry of Transport, Htun Lwin Oo, personal
communication, 2015). Most likely, water demand in Myanmar will increase in the future due to
enhanced production and trade in agricultural products, the expansion of transport systems via rivers
and ports, and the anticipated growth of cities and industries (The World Bank, 2014). This increasing
water demand and the high rainfall variability in the dry zone will probably cause the construction of
more pumping stations for both groundwater and river water as well as the building of more reservoirs
and dams.
2.5.3    Cyclones
The coast and the delta zones of the Ayeyarwady and Chindwin River are extremely exposed to
impacts from cyclones associated with winds, storm surges and salt water intrusion into groundwater
(Rao et al., 2013). The Ayeyarwady Division is, compared to other regions in Myanmar, densely
populated (177/km²; Department of Population, 2015)) and the extensive and shallow continental shelf
of the Andaman Sea allows cyclones and storm surges to inundate the delta and some inbound areas
(Webster, 2008). Tropical cyclone formation in the northern Indian Ocean occur preferentially before
(April-May) and directly after (October-November) the Asian summer monsoon season (Webster,
2008). During the cyclone Nargis in the year 2008, which was the most devastating cyclone to strike
Asia since 1991, the Ayeyarwady River delta region was flooded by a 3.5 meter wall of water
(Thomson Reuters, 2009). Wind speed was in excess of 65 ms-1 (Webster, 2008). More than 130,000
people died and 2.4 mio people were severely affected (van Driel and Nauta, 2013; Thomson Reuters,
2009). Nargis caused severe harm to the winter rice crop and loss of rice seed and Myanmar faced
food shortages after the event (Webster, 2008). Seawater inundated large areas of the Ayeyarwady
delta posing challenges to future rice production (Webster, 2008). Lin et al. (2009) detected a pre-

existing warm ocean anomaly in the Bay of Bengal which was probably the cause why a weak

category-1 storm could rapidly intensify to an intense category-4 storm within only 24 hours.

Mangrove clearance for shrimp farms and rice paddies was probably a major factor in aggravating the

impacts of cyclone Nargis (Nature News, 2008). Historically seen, Myanmar has only infrequent

tropical cyclone landfalls but since 2006, there has been an apparent increased activity in the Indian

Ocean. Whether this development is part of a continuing trend due to climate change is difficult to

assess because data quality and length of the records are limited (Webster, 2008).

## 2.6    Flora and fauna

Myanmar is one of the few countries in Southeast Asia with relatively high levels of biodiversity and

intact forest areas (Rao et al., 2013). About 48%, or 317,730 km² of Myanmar's surface is covered

with closed tropical forest; however, according to the FAO, both quantity and quality are decreasing

(Htun, 2009). In the early 1990s, Myanmar had still a total forest cover of about 442 000 km², which is

67% of the total surface area (Leimgruber et al., 2005). The forest flora ranges from sub-alpine to

tropical formations (Aung, 2002). The forest along the Salween River on the Thai-Burmese border lies

on a bio-geographic border that is rich in biodiversity, in wildlife and fish populations, and this area is

one of the most fertile areas for teak in the world (Salween Watch and SEARIN, 2004). Tropical

evergreen rainforests occur in areas receiving >2,000 mm of rain annually and they are home to many

birds species. Many wild animals which were once plentiful, are now reduced in number and are

protected, e.g. the 'Irrawaddy dolphin', the Asian two-horned rhinoceros, the wild water buffalo, the

gaur and other deer species (Hadden, 2008; Smith et al., 2009; Aung et al., 2013).

All species play an important role in maintaining balance in and supporting ecosystems. If these

significant values and benefits are lost, humans will response with additional inputs to maintain the

system's functionality (Allen et al., 2010). The majority of threats to Myanmar's biodiversity are in

general linked to human population growth and economic development, and the corresponding

increasing demand for natural resources and space (Allen et al., 2010). Overexploitation of fishes is a

major concern for the country's inland fisheries which are likely to increase due to political and

economic transitions (Rao et al., 2013). However, little is known about species-ecosystem interactions

to be sure of human (e.g. dam projects, mining) or climate impacts (e.g. temperature changes may lead

to alien species invasions). Following Allen et al. (2010), alien species invasions, pollution from

mining activities, river flow modifications and overexploitation of fishes are the major threats to the

biodiversity of freshwater systems in Myanmar.

## 3 Social and economic features

### 3.1 Agricultural land use

Agriculture is the main pillar of the country's economy and contributes ~37% to the GDP (Ministry of Forestry, 2005; CIA, 2015). The estimated cultivated area in Myanmar is 18.27 million ha which is equivalent to 55 % of the cultivable area (FAO, 2015). More than 65% of the population live in rural areas, working in the agricultural sector (FAO, 2014). The major agricultural products are rice, pulses, beans, sesame, groundnuts, sugarcane and hardwood. 42% of Myanmar's cropland is cultivated with paddy rice, particularly in the Ayeyarwady delta region (FAO, 2004). The delta areas and river mouths are the most populated sections within the river basins. Here, cultivation of rice in flooded paddies predominates (FAO, 2004). In general, the agricultural practices are still very low tech, and usually water buffalos are used for ploughing (van Driel and Nauta, 2013). The majority of the farmers there are small-scale landholders with an average lot size of 2.27 ha cultivating paddy fields during the monsoon season and vegetable gardens on the river banks in the dry season (Salween Watch and SEARIN, 2004). All-the-year, they cast for fish in the rivers and along the coasts. The country has the largest estimated population of small-scale fisheries in the world (SEAFDEC, 2012). The government is the ultimate owner of all land in Myanmar and the farmers are only allowed to cultivate the land with the government's prescription. One third of the rural residents are landless labourers (Hiebert, 2012). Land-grabbing and confiscation by the military, government and international investors are a huge problem, particularly in the Tanintharyi Region, followed by Kachin State (Farmlandgrab, 2014).

The mangrove forests in the delta and coastal areas supply firework and bark for tanning which has already led to critical degradation of the ecologically important mangrove forests (Webb et al., 2014). The Ministry of Forestry in Myanmar (2005) estimated that the mangrove forest area decreased to about almost half of its size between 1990 and 2002. This development is likely going on due to the increasing number of fish and prawn ponds, salt evaporation ponds for commercial purposes and the expansion of agriculture land for food security (Ministry of Forestry, 2005).

Following categories of farmland exist in the country (JICA 2013, p.9): 1. Paddy field or wet land which can be used for paddy farming (so called Le), 2. Upland farming (Yar), 3. Farmland which appears in the floodplain in the Ayeyarwady River as the water recedes (Kaing), and 4. Farmland which appears on the sandbars in the Ayeyarwady River as the water recedes (Kyun) (Fig.2). Farming on flood plains and sandbars of the Ayeyarwady River is of interest due to the relatively good conditions of fertility and access to water for irrigation either directly from the river or from shallow groundwater aquifers. In contrast to the rainfed upland farms, where the groundwater aquifer is drawn out by tube-wells, exploitation of water for irrigation is much easier and less costly.

We identified via remote sense analysis that in 2016 roughly 8 % of the area in the central dry zone is
alluvial farming land. The amount of farmland used for alluvial farming increased slightly from 1988
(3,855 km²) to 2016 (5,511 km²) from 5,6 to 8% of the total farmland. The alluvial land can be used as
farmland only during and after the raining season, thus there is only a short cultivation period.
About 22% of the annual paddy production of Myanmar is generated within the central dry zone
(McCartney et al., 2013). Furthermore, 89% of Myanmar's sesame production, 69% of the groundnut
production and 70% of the country's sunflower production are generated within this area (McCartney
et al., 2013). Pulses and cotton are other important crops in this region.
3.2    Water use and management
Myanmar has abundant water resources including both surface and groundwater. The potential water
resources volume is estimated to be about 1,000 km³ for surface water and about 500 km³ for
groundwater (WEPA, 2014; Oo, 2015). The country's total renewable water resources are 24,352
m³/year per inhabitant but only 5% of its physical water resources are used at present (WEPA, 2014).
Water utilization for the agricultural sector is about 90% while industry and domestic use is only about
10% of the total water use. Due to ongoing and expected future economic development and population
growth, it is obvious that the physical potential for further development of water resources is
substantial (WEPA, 2014).
Several national ministerial departments are responsible for the coordination of water-related issues in
Myanmar. There is the Department of Irrigation, the Water Resources Utilisation Department, the
Ministry of Rural Development (domestic water), the Ministry of Environmental Conservation and
Forestry (MOECAF), and the Department of Meteorology and Hydrology and the Directorate of
Water Resources and River Improvement, both associated with the Ministry of Transport.
*Central dry zone*
Farming in the central dry zone is only possible with irrigation due to the high variability of rain falls.
Irrigation in the dry zone has its beginning in the 11th century when weirs and tanks were constructed.
The first groundwater and surface water pumped systems were initiated in 1962 and they significantly
contribute to increased food security in the central dry zone (McCartney et al., 2013). The annual
recharge of groundwater in the dry zone is estimated around 4,770 Mm³ and the annual total use is
>770 Mm³ (data from 2000; Johnston et al., 2013). In this region, irrigation is mainly conducted by
canal systems from the rivers to the arable land while groundwater withdrawal still plays a minor role.
However, the number of pumping systems is increasing, particularly through Chinese investments
(Johnston, R., 2015, personal communication). Rainwater harvesting and storage is another simple and
common method for domestic and livestock purposes in the villages. During the dry season, village
ponds dry out frequently. This problem is often solved by groundwater or river water pumping to the
ponds (Johnston et al., 2013), which is in some regions conducted by the local government who sells
the water to the villagers (personal communication from a resident in Bagan, 2016).
*Ayeyarwady delta area*
Embankments, sluice gates and drainage systems have been constructed to protect the agricultural land
in the lower delta against saltwater intrusion (van Driel and Nauta, 2013). During the monsoon season,
rainwater is stored in drainage canals for the dry period. The gates of the sluices are kept open from
mid-May to mid-September in order to control the water level of the drainage canal. Old river courses
are functioning as major drainage canals but there are also smaller artificial drainage channels (van
Direl and Nauta, 2013). Although these drainage systems are quite proven for a long time, intrusion of
saline water is a major concern in this area because of leakages, dam failures or natural hazards such
as storm surges and cyclones. During the dry season, irrigation is practiced in the delta by pumping the
water from the channels to the paddy fields. In the middle part of the delta, tidal irrigation is
extensively practiced and possible due to sufficient flow of river water to the ocean (van Driel and
Nauta, 2013).
3.3    Hydropower and river flow modifications
Myanmar's major rivers are still less regulated compared to other Asian rivers (Hedley et al., 2010).
There are currently no dams on the mainstream of the Ayeyarwady River. However, about 1,300 km
of embankments were built during the late nineteenth and early twentieth century (Hedley et al.,
2010). Between 1988 and 2003, the government of Myanmar has constructed about 150 smaller dams
and reservoirs and 265 river water pumping stations along the tributaries (Ministry of Forestry, 2005).
The Ayeyarwady River is subject to numerous potential dam projects and seven dams are currently in
the planning stage (Allen et al., 2010). Several dams are also planned along the Salween River which
likely will impact both the hydrodynamic and the sediment load (Salmivaara et al., 2013). In 2011,
planned hydropower dam constructions by the China Power Investment Corporation near Myitsone at
the confluence of the Mali and the N'Mai Rivers (Fig.2) were halted due to peaceful public protests as
well as armed resistance (Burma Rivers Network, 2014). The dam was intended to build 152 m high
and it was envisaged to inundate 47 villages and to displace ca. 10,000 people in the Kachin State
(Burma Rivers Network, 2014). Another critical point is that the northern part of the country is prone
to earthquakes and a broken dam would have catastrophic impacts on downstream areas and the city of
Myitkyina, the capital of the Kachin State (Burma Rivers Network, 2014). It is expected that building
larger dams will come along with social impacts like displacements, food security, health concerns,
and the loss of culture (Smakhtin and Anputhas, 2006; Burma Rivers Network, 2014). Myanmar has
experienced a rapid growth of hydropower capacity with a potential of almost 40,000 MW, of which
only 6% have been developed. Hydropower supplies the majority of the electric exports supported by
foreign investments (ADB, 2012; Kattelus et al., 2014).
River flow modifications lead to changes in the composition and diversity of aquatic communities.
Aquatic species have evolved life history strategies primarily in response to the natural flow regimes.
Therefore, flow regime alterations can lead to loss of biodiversity of native species (Smakhtin and
Anputhas, 2006). Dam building results in a range of upstream and downstream impacts, not least
disruption of migratory routes and of breeding patterns (Nilsson et al., 2005). Water abstraction and
damming are one of the major threats to freshwater biodiversity (Allen et al., 2010). In the deltas,
mangrove forests rely on the non-saline water from rivers. Any reduction in the volume of sweet-water
to their roots causes mangroves to dry up, resulting in salt-water intrusion, and subsequent soil-
erosion. It is further assumed that the construction of dams would accelerate the deforestation in the
Salween River basin, with severe negative effects on biodiversity and the dense dry deciduous forests
also called teak forest, which is crucial for the livelihood function of local ethnic people (Salween
Watch and SEARIN, 2014). In general, the full scope and scales of potential environmental and
ecological impacts from dams is largely uncertain due to the complexity of feedback mechanisms and
system response (Fan et al., 2015), particularly in regions where the rivers play such an important role
like in Myanmar. Dams will alter the river flows as well as the sediment load, which will impact the
further development of the Ayeyarwady delta. For the navigability of the rivers and the canals, a
decrease of the sediment load would be a favourable effect of dam building.
China has an increasing interest in covering its energy demand, forced by the international community
to get out of $CO_2$-emission intensive power generation. Making investments in hydropower in
Myanmar in order to provide energy for the western part of China solve these challenges for now. At
first glance, both nations benefit from this energy trade. Building dams could potentially increase the
irrigation opportunities, particularly in the central dry zone of Myanmar. It would enhance navigation
possibilities and provide flood control (Lu et al., 2014). On the one hand, the energy trade is an
economic and political opportunity because it must be based on cooperation between Myanmar and its
neighbouring countries and counters the isolation status which is partly still existent (Kattelus et al.,
2015). On the other hand, damming Myanmar's rivers could have very serious negative effects on the
river biodiversity and the stability of the deltas (Hedley et al., 2010). A decreasing supply of the fertile
alluvial sediments would modify the availability of agricultural land in an unknown dimension. It is
expected that deforestation would further increase in the dam building areas as a result of
infrastructure plans, with severe impacts on local biodiversity, local people, hydrology and on regional
and even global climate.
India, Bangladesh, China and Thailand have different interests in Myanmar's water resources and all
of them are involved in diverse hydropower project plans. These natural resources as well as
Myanmar's convenient geographical and strategic geopolitical location will possibly strengthen the
country's economic and politic role in Southeast Asia. Negative aspects of hydropower development
are the risk of rising conflicts between ethnic minorities and the military (Burma Rivers Network,
2014) and also between Myanmar and neighbouring countries due to differing interest and needs of
the water resources.
3.4    River ecology protection
All aspects of water resources conservation are unified in the Conservation of Water Resources and
Rivers Law, enacted in 2006. It aims to conserve and protect all water resources and river systems for
beneficial utilization by the public, to protect the environment, to smooth and safety waterways
navigation along rivers and creeks and to contribute to the development of State economy through
improving water resources and the river system (The Union of Myanmar, 2006). Mining within 100 m
of the Ayeyarwady, the Salween, the Chindwin and the Sittaung rivers is banned by the Ministry of
Mines (Schmidt, 2012). However, despite these ambitious laws, freshwater diversity, including inland
wetlands, estuaries and mangroves, appear to be limitedly protected in Myanmar (Salmivaara et al.,
515 2013).

In 2013, a National Water Resources Committee (NWRC) has been established by a Presidential
decree. The NWRC stated that the weak cooperation between the water-related agencies in Myanmar
is the major problem (Win, 2014). The committee follows the vision "*In 2020 Myanmar will become*
*water efficient nation with well developed and sustainable water resources based on fully functional*
*integrated water resources management system*" (Win, 2014). The NWRC concludes that more
research is needed to solve the problems in Myanmar's river basins (Win, 2014).
**4    Climate change impacts and future perspectives**
Only very few studies on climate change impact assessments in Myanmar have been conducted so far
(Shrestha et al., 2014). During the past decades, inter-seasonal, interannual and spatial variability in
rainfall has been observed across all Southeast Asian countries (IPCC, 2014). However, detailed
studies for Myanmar in particular are lacking, but a similar pattern can be assumed due to the
influence of the same monsoonal atmospheric circulation system. A substantial inter-decadal
variability exists in the Indian monsoon circulation which is particularly crucial for the central dry
zone (IPCC, 2014). Extreme weather events have become more frequent and intense during the last
decades related to their direct impacts on socio-economy what could also be detected for Myanmar
(GCCA, 2012). Most likely, the intensity and frequency of droughts in the dry zone particularly during
ENSO events will increase (IPCC, 2014). Variability of river runoff and changes in seasonality are
expected for Southeast Asia as a result of climate change (IPCC, 2014). Sea level rise, decreasing river
runoff and increasing intensity and frequency of droughts will lead to even more increased saltwater
intrusion into river deltas. In the medium term, enhanced glacier and snow melt in the source areas of
rivers will cause generally higher discharges and potential floods. However, individual glaciers are
currently advancing or stable in Asia depending on their particular features (Scherer et al., 2011).
Studies on the glaciers feeding the Ayeyarwady have not been conducted yet. The low-lying
Ayeyarwady delta is particularly exposed to sea-level rise and vulnerable due to its high food
productivity and population density. It is assumed that a 0.5 m sea-level rise would advance the
shoreline along the Ayeyarwady delta by 10 km inland (NAPA, 2012). Changes in river flow will
likely increase the risk of flash floods and lowland regions will be regularly inundated (NAPA, 2012).
Furuichi et al. (2009) showed a decrease of the annual discharge of the Ayeyarwady River over the
last 100 years based on a statistical comparison with data collected in the 19th century, but the driving
forces remain unclear. The central dry zone experienced higher maximum temperatures and lesser
rainfall in the 1990s compared to other regions in Myanmar (Ministry of Forestry, 2005). This is
hypothesized as a result of anthropogenic climate change and global warming (Ministry of Forestry,
2005).  Increasing temperatures in this region will raise the concentration of dissolved salts in the
ponds, channels and other storage systems resulting in a reduction of drinking water (NAPA, 2012).
Climate change is expected to exacerbate existing threats to biodiversity in Myanmar through its
impacts on humans and their dependence on products and services produced by freshwater ecosystems
(Rao et al., 2013). Changes of rainfall regimes, air and water temperature and evapotranspiration will
affect distribution and abundance of freshwater species in unknown ways (Rao et al., 2013).
Particularly the Ayeyarwady River basin will most likely be affected by population growth,
urbanization, land use change and climate change in the future (Bates et al., 2008; Salmivaara et al.,
2013). Rao et al. (2013) concluded, based on findings from Iwamura et al. (2010), that the
Ayeyarwady dry forest located in the central river basin is particularly prone to future changing
rainfall and temperature conditions. The authors expect that the seasonal amount of rainfall will
decrease which will exacerbate the already water-stressed region (Rao et al., 2013).
Continuing loss of natural forest cover and mangrove habitats can influence processes affecting
climate change by release of $CO_2$ to the atmosphere (Van der Werf et al., 2009). It can be summarized
that climate change most likely will impact the river basin ecosystems in Myanmar in a so far
unknown dimension through modification of seasonal flow regimes and the timing, extent and
duration of floods and droughts. Climate change projections and scenarios have not yet been
developed for Myanmar in particular. There are numerous assumptions and expectations but no
detailed data for the country. This lack of future assessments is also a result of the nonexistence of
paleoclimate data.
Due to the lack of scientific research in the country, often uncertain or incomplete data bases and rapid
political and economic changes, future perspectives for human-water dynamics in Myanmar's river
basins can only be assessed with high uncertainties. However, it should be possible to indicate the
major drivers of future changes. Undoubtedly, the availability and quality of freshwater is and will be
the core of the country's future development but increasing conflicts on water may arise due to
growing foreign investments and various international and national interests.
Findings from Salmivaara et al. (2013) indicate that the Ayeyarwady delta, the Salween river mouth
and the central lowlands in the Ayeyarwady River basin are under the highest pressure as a result of
intensive land use, high population density and vulnerability to water pollution. These regions are
most likely to be exposed to further pressures such as urbanization, land use change and climate
change (Bates et al., 2008). Major challenges for the Salween river basin will be linked to extensive
dam projects (Burma River Networks, 2014; Kattelus et al., 2014). The major challenges for Myanmar
are seen in covering the balance between national societal, economic and political development and
the urgent need to protect and conserve its water resources and biodiversity.
**5    Selection and identification of human-water dynamic key indicators**
Human-water dynamics include one-way causal chains as well as complex feedback mechanisms.
Particularly in a country like Myanmar where water plays such a major role in people's life, detailed
knowledge and understanding of human-water interactions is essential in order to evaluate possible
future developments. This knowledge is crucial for a proper and sustainable water management that
meets the social, the environmental, economic and the political interests. Our first step for future
human-water dynamic studies in Myanmar is therefore the selection and identification of
environmental key indicators based on the reviewed literature within the here presented paper and
based on own observations during field studies. Environmental key indicators provide information on
complex issues in a simplified manner and characterize major causal impact-response chains. They
can be used for future development assessments and current state analyses.
For the here presented study, the eDPSIR (enhanced driving force-pressure-state-impact-response)
concept  by Niemeijer & de Groot (2008a, b) is seen as a suitable framework to structure the selection
of relevant environmental indicators. This framework is an enhancement of the DPSIR (driving force-
pressure-state-impact-response) approach which has been applied to several water-related
environmental studies in order to identify causal chains (e.g. Pirrone et al., 2005; Kagalou et al., 2012;
Pinto et al., 2013; Geng et al., 2014). The advantage of an enhanced DPSIR application is that this
framework is causal network based and includes the interrelations and feedbacks between various
causal chains within a system. First, we follow the steps to build a causal network proposed by
Niemeijer & de Groot (2008):
*Step 1: Broadly define the domain of interest:* Human-water interactions in Myanmar's river basins
*Step 2: Determine boundary conditions:* Socio-hydrological system in the humid tropics, monsoon
influenced
*Step 3: Determine the boundaries of the system:* In situ situation in the river basins with particular
focus on the Ayeyarwady River basin
*Step 4: Identification of abstract indicators for the main factors and processes:* (Table 1). Energy
needs, land use intensification, increase of atmospheric $CO_2$, global warming, expansion of industrial
zones and demand for wood are examples for driving forces in Myanmar's river basins. These drivers
create pressures which in turn modify the state of e.g. river discharge, soil degradation, water quality,
and so on. Changes in the state of e.g. water quality impact aquatic biodiversity and the availability of
drinking water. The last row, responses, has been omitted since this aspect is not in the focus of this
paper and particular responses of the society or government have to be studied in the future more in
detail.
*Step 5: Iteratively mapping the indicators in a direction graph*: Fig. 3 shows a causal network of
selective indicators for human-water interactions in Myanmar. There is no claim for completeness
regarding the specific links and feedbacks. It is a first attempt to structure the relationships between
and within water-related social and physical-environmental indicators in Myanmar's river basins.
Fig. 3 demonstrates the complexity of a causal network of indicators for human-water dynamics in
Myanmar. Mapping this network helps to identify important nodes and to structure further study
approaches. Runoff for example seems to be an important end-of-chain node (cf. Niemeijer & de
Groot, 2008a, b) as well as fish population, which is indicated by many incoming arrows, whereas
dam and reservoir building and deforestation represent a central node with several incoming and
outgoing arrows. It is challenging to identify a typical root-node indicated by many outgoing arrows.
Climate change might be a root-node within this network. It is undoubtedly triggered by human
activities, though rather on a larger spatial scale and the impact of the Burmese people on global
climate change is comparatively small at least at the current state.
Furthermore, Fig.3 clearly exhibits that studying human-water interactions essentially need the input
from social as well as from natural science and it is indispensable that experts from both disciplines
exchange their knowledge and work together on the same research questions.
*Alluvial framing as an example for human-water dynamics within the Ayeyarwady River basin*
Alluvial farming can be seen as a demonstrative example for human-water dynamics in the dry zone of
Myanmar. Assuming that climate variability in terms of (monsoon) precipitation variability is one of
the root-node indicators of human-water dynamics in Myanmar, changes of precipitation amounts or
timing cause high hydromorphological and sedimentation dynamics in the Ayeyarwady River. Lower
rainfall amounts or dry periods result in lower river discharge and foster the accumulation of sandbars
in the river bed. Moreover, land use changes and forest logging have an additional influence on
sedimentation loads in the river and creates new fertile floodplains. Most likely, these processes have a
visible impact on alluvial farming in the dry zone because more fertile arable land with good access to
irrigation water is available. This is of even higher importance in the light of increase of dry spells and
changed timing of the monsoon rain in the dry zone. Our remote sensing analysis shows an increase of
alluvial farming by 1,656 km² (almost 70%) since 1988. Most of the alluvial farmers grow crops like
onions because of market prices, suitability to alluvial land, and short term benefits (personal
communication with citizen of the dry zone). Concurrently small scale alluvial farming implies a
potential higher flood risk and related crop failure and loss of yields for the farmers and livelihoods of
their families and communities. However, if and how these observations are actually an impact-chain ,
has to be investigated and is subject for further research.

## 6     Conclusions

Myanmar's economy and the people's income and well-being strongly depend on the quality and
availability of sufficient water resources. The delta region of the Ayeyarwady River and the central
dry zone are the areas most populated and most intensely used by agriculture in the country. On the
one hand, the farmers depend on frequent river flood events because the river provides fertile alluvial
soils; on the other hand, they suffer from water-related extreme events such as floods or drought
periods. It is expected that theses climatic extreme events will likely increase in frequency and
magnitude in the future as a result of climate change. Different national and international interests in
the abundant water resources may provide opportunities and risks at the same time for Myanmar.
Several dam projects along the main courses of the major rivers are currently in the planning phase.
Dams will most likely modify the river flows, the sediment loads and also the still rich biodiversity in
the river basins, in a still unknown dimension. On the other hand, these foreign investments allow the
development of infrastructure and probably stabilize the political relations between Myanmar and its
neighbouring countries and strengthen its role in Southeast Asia and even globally.
All authors of the reviewed literature agree that Myanmar is facing big water-related challenges.
However, future perspectives and developments are mostly still intangible due to the large gap of
research in the country and the limited detailed knowledge about the status-quo. More in-depth
qualitative and especially quantitative analyses on human and climate impacts on Myanmar's water
resources are strongly required in order to adapt water and land management to current and future
climate change. The year 2008 was a kind of turning point when cyclone *Nargis* made landfall in the
Ayeyarwady delta region. Since then, a number of action plans have been established with the aim to
call attention on extreme weather events. Furthermore, the vulnerability of the Burmese people is
increasing because population pressure is forcing more people to live and work in coastal zones and
river basins. The central dry zone and the delta zone are the most vulnerable parts of the river basin
related to climate change and also to human impact.
Relevant aspects for future development of Myanmar's river basins combine environment-water-
related indicators, climate, economic and social development, water management and land use
changes. Research on this interplays need to capture the spatial and temporal dynamics of this drivers.
Yet, it is only possible to gain a full understanding of all these complex interrelationships, when multi-
scale spatiotemporal information will be analysed in an inter- and transdisciplinary approach. The
eDPSIR approach is considered to be a suitable starting point for human-water research in Myanmar.
The here presented indicator scheme (Fig.3) was a first attempt to structure the reviewed information
and to provide a first assessment on relevant indicators and key nodes on the socio-economic as well
as on the physical side. Alluvial farming is a suitable example for human-water dynamics in
Myanmar's central dry zone and it could be demonstrated that the share of alluvial farmland increased
by 70% since 1988 in this region. We hypothesize that the increase of alluvial farming is an effect of
hydromorphological impacts  (potentially enhanced by human interventions like forest clearing).
Concurrently this land is largely used because of relatively good farming conditions (fertile soils and
good access to water for irrigation) compared to upland farmland in the central dry zone especially in
light of the difficult climate conditions. However, increased alluvial farming increases the potential
flood risk for the farmers' livelihoods. Yet, this hypothesis has to be investigated in further studies.

**Acknowledgements**
The authors are grateful to Stephan Opitz (University of Cologne) and Henrik Bours (University of
Bonn), whose critical comments helped to improve the manuscript. Adrian Almoradie (University of
Bonn) assisted the preparation of the overview map. Furthermore, we thank Mukesh Boori (Samara
State Aerospace University Russia) for the remote sensing analysis on alluvial farmland. The research
was funded by the German Science Foundation (DFG). The authors would like to thank three
anonymous reviewers for their useful comments on earlier versions of this manuscript.

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

**Figure 1**

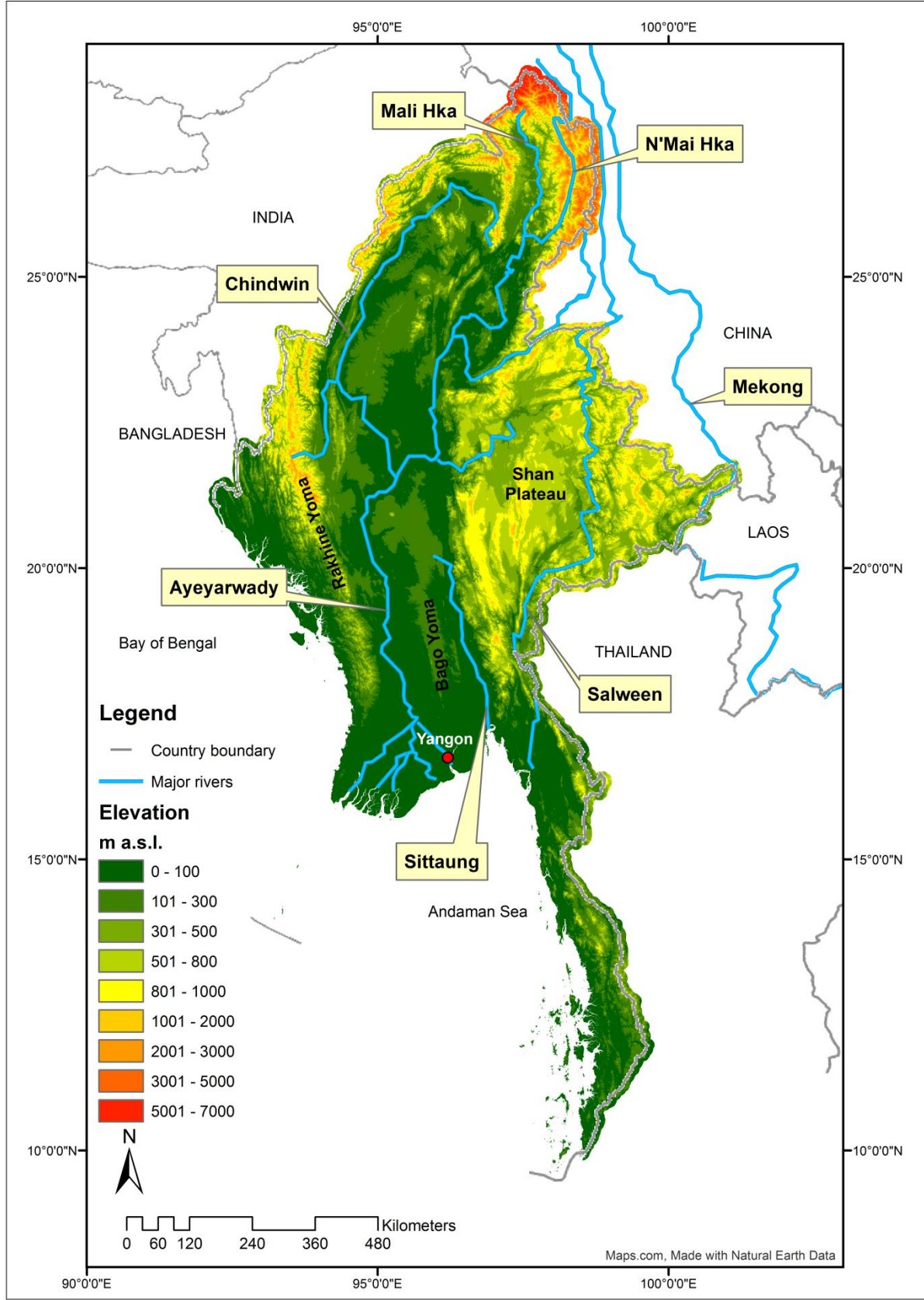

**Figure 2**

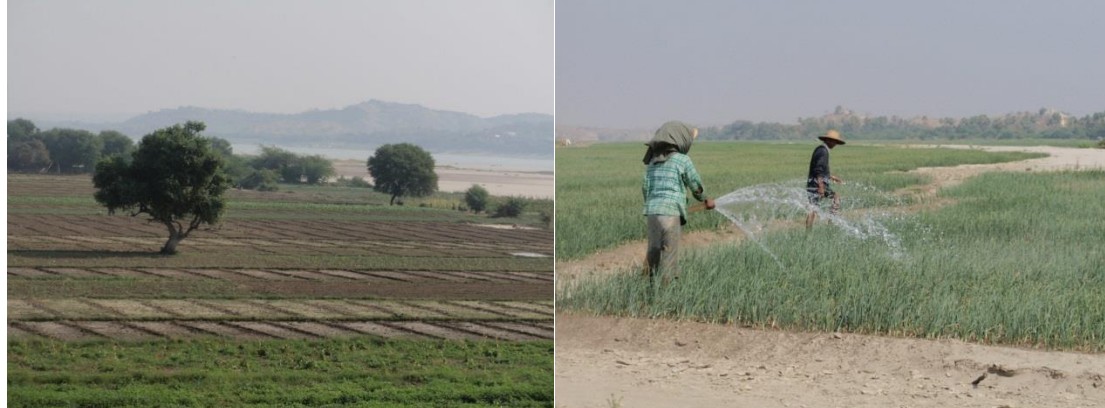

**Figure 3**

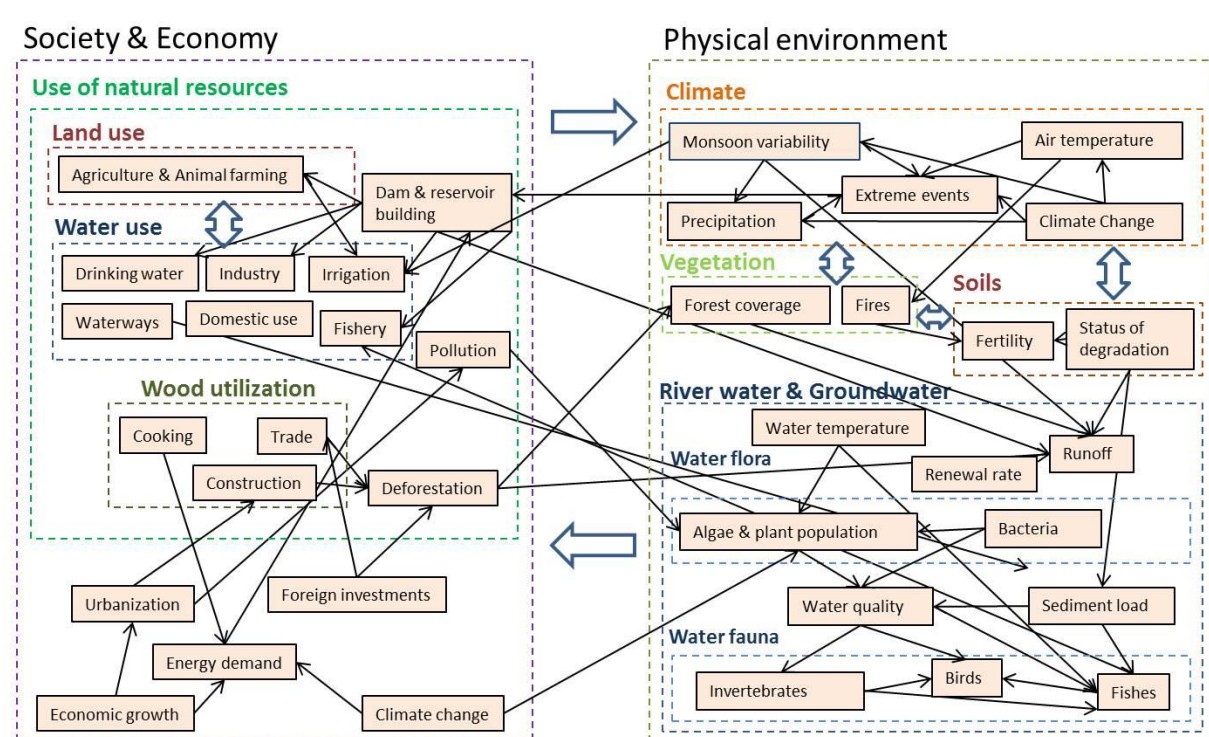






**Figure captions**

Fig.1: Physical overview map of Myanmar including state border lines, major rivers and mountain ranges.

Fig.2: Left:. Example of Kaing farming (farmlands which appear in the flood plain in Ayeyarwady River) in Magway district (Chauk). Right: Example of Kyun farming (farmlands which appear on sandbars in Ayeyarwady River) in Pakkoku township. Pictures taken by M. Evers

Fig. 3: A causal network to demonstrate specific (thin black arrows) and general (large blue arrows) links between and within water-related social and physical-environmental indicators in Myanmar's river basins.

Table 1: Selection of general indicators within the DPSIR framework. Respective responses are not listed because they are subject of our future research and will be studied more in detail.

| Driving force | Energy demand | Land use intensification | Increase of atmospheric $CO_2$ | | Industrialisation | Demand for wood /wood trade |
|---|---|---|---|---|---|---|
| Pressure | Building hydropower dams | Increase of water withdrawal and groundwater pumping | Increase of temperature and evaporation | | Polluted sewage release | Deforestation |
| State | Change of river flow | Decrease of groundwater level | Change of precipitation (monsoon) patterns | Increase of glacier melt | Deteriorating of water quality | Soil degradation |
| Impact | Biodiversity  Fish migration | Shortage of groundwater | Longer dry periods, droughts, higher maximum temperatures | Seasonal shift in river discharge  Agriculture, biodiversity | Availability of water in good quality for humans and agricultural use, biodiversity | Increase of erosion processes and sediment load in the rivers |