# Peer review of "A review of current and possible future human-water interactions in Myanmar's river basins"

_Hydrology and Earth System Sciences, 2015_

## Referee Comment (RC1) · Anonymous Referee #1 · 16 Jan 2016

In this paper the authors are anticipating the nature of coupled human-water system interactions and feedbacks that could be expected in Myanmar's river basins, with the focus on those in the Ayeyarwady delta. They have attempted to do this using a framework for socio-hydrology proposed by Sivapalan et al. (2014).

I do welcome this attempt as I think this kind of long term (decade to century scale) analysis will be able to embrace the suite of climatic, hydrological, socio-economic and socio-cultural elements in projections in the future. After reading the manuscript, I can indeed see the place of socio-hydrology in such long term projections, including especially the two-way feedbacks between humans and water in these river basins. Since these river basins are rather large, and considering the heterogeneity of the hydro-climatic situation as well as the diversity of human populations and occupations, there is also a spatial dimension to the socio-hydrology.
[Figure]

In spite of my enthusiasm for the paper, I am not sure that the current manuscript can stand alone as a research article. At best it is a literature review or weakly formulated opinion article. I would like to see the idea developed some more, before it can be published.

I understand that the idea is at a concept stage, not even a proposal stage, and there are no research results to present. Still I would like the authors to take the idea further, into at least a proposal stage. In two recent socio-hydrology papers, we have seen the recommendation to do "framing" of a socio-hydrologic problem. Framing involves using available information to identify a phenomenon and the domain/scale, putting together a perceptual model, choosing state variables, causal factors that affect the state variables, developing functional relationships, estimating parameters, and finally model parameter estimation. At the very least the authors could start with starting with a narrative and progressing to the stage of developing a perceptual model (and some hypotheses for identified phenomena), then it will qualify to be a publishable research article. It is disappointing that the paper has only two figures, one borrowed from Sivapalan et al. (2014) and the other a map of Myanmar. There needs to be more substance.

Therefore, this calls for major revision of the paper.

For more ideas on "framing" I encourage the authors to look up two recent papers (and references found therein).

Sivapalan, M., and G. Blöschl (2015), Time scale interactions and the coevolution of humans and water, Water Resour. Res., 51, 6988–7022, doi:10.1002/2015WR017896.

Garcia, M., K. Portney, and S. Islam (2016). A question driven socio-hydrological modeling process. Hydrol. Earth Syst. Sci., 20, 73–92, doi:10.5194/hess-20-73-2016

---

## Referee Comment (RC2) · Anonymous Referee #2 · 15 Feb 2016

The paper describes human and water systems in Myanmar. The topic is relevant and, given the focus on a very dynamic system, I think that this paper has a great potential. Yet, while the title suggests a focus on human-water interactions, e.g. socio-hydrology, this review is mainly a description of either natural or social processes. Very little is said about the reciprocal effects, mutual interactions and feedback loops between nature and society. I would not recommend its publication on HESS because I see little value of the current manuscript from a scientific viewpoint. For instance, the authors often quote Sivapalan et al. (2014) and they also report their conceptual framework (Figure 1). In the paper, however, there is no real attempt to follow it and show it with reference to Myanmar. Section 2, 3 and 5 do follow the three elements of the feedback loop (with in between a "out-of-the-blue" section 4 about flora and fauna), but they do not describe how each element impacts or responds to changes of the

other elements, i.e. human-water interactions. In other words, the arrows of Figure 1 (which are, in my opinion, the essence of the referred paper of Sivapalan et al., 2014) are not sufficiently reflected into the text. Addressing this point (i.e. describing the interplay of the three elements, rather than just the three elements) can be a way out to make this paper scientifically interesting and potentially publishable. More minor points: 1) Reference to recent literature about human-water interactions is also very limited, see e.g. 2015 WRR debate, tons of HESS papers on the subject and recent socio-hydrological studies about Bangladesh, a neighbouring country, China and Australia. 2) English should be improved, as there are numerous typos. 3) I have some more technical comments that might follow if major concern is addressed.

---

## Short Comment (SC1) · 24 Feb 2016

Dear anonymous reviewer,

thanks a lot for your productive critizism. You are right with your comments and I just would like to add, that we are working on a socio-hydrologic frame of our paper since the first review focus on similar aspects. It is quite difficult to find reliable data for the country of Myanmar, however, we're going to write a new version of the text in which we try to match the real-world data to the socio-hydrologic concept.

---

## Editor Comment (EC1) · T.A. Bogaard (Editor) · 30 Mar 2016

**Comments**

This manuscript addresses an important issue on human-water interactions in Myanmar's river basins. Authors' efforts are highly appreciated. However, the manuscript is found to be merely a review which is too much dependent only on the literature cited. Some of the references are not updated ones, and the authors did not seem to make a new analysis at least either based on available information or on their own survey. There are a lot of shortcomings with respect to the review of current water resources sectors and related ones. The manuscript needs a thorough revision as a whole, and some validations, if possible based on own survey, are also necessary.

Of course, I agree with that there are very few research publications about every sector of Myanmar, and it is also very difficult to get access to the published documents many of which are only available within the country. Consequently, the discussion could result a departure from the existing conditions in the country. Moreover, scarcity on reference information is not an excuse for a research which could be published to world-wide readers.

I **do not recommend** the manuscript is suitable for further publication in its present form.

**Specific comments:**

1. In Abstract, what does the "sound knowledge" mean for? It must be clarified.
2. In abstract, line 9-12. The meaning of the clause is not clear. "Though" should be omitted in the sentence.
3. In Abstract, line 17-19. "On the one hand" must be changed to "On one hand".
4. In the introduction, line 16. "Burmese" must be changed to "Myanmar". Like this other places must be changed.
5. Page 4,line 21. Source citation should not be used such as BBC News.
6. Page 5, line 21-23. The clause "We hypothesize…. Permanently change" is not clear. What are the all components? Are they interacting with each other? Which driving forces do authors mean? Please provide examples (such as regional climate change?).
7. Page 6, line 19. Myanmar is not the largest country in Southeast Asia with respect to overall country's area, and it is the second largest country after Indonesia which comprises of many large islands. However, considering the size of successive land, Myanmar owns the largest land size in Southeast Asia. Therefore, the authors must provide the clear information to readers. Since the sentence (Page 6, line: 18-21) has no citation, and therefore it must be authors' finding or knowing via some calculations or using existing maps. Please make sure what the real message is for the readers.
8. Fig. 2. The quality of the figure is not good. The legends for administration boundary and state boundary are not clear. Different colors for these legends should be used. Moreover, only three major cities are given on the map, pointing with red circles, but the legends are not provided. If it is not necessary, they can be omitted. If not, please provide necessary

legends for the major cities mentioned on the map.Overall, the quality of the figure can be improved by GIS techniques, providing clear information, labels and associated legends.

9. Section 2.1. Page 6. How do you decide the important physical features? Only two features are given, i.e. physiographic characteristics and soil types which are retold from previous studies. How about others, such as geology?

10. Page 8, line 4-13. Five sub-climate regions for Myanmar are mentioned according to Koeppen-Geiger classification. It is better to provide a figure showing the country's map with these five sub-climate boundaries.

11. Page 9. Line 20-22. The citation is too general. El Nino events not only results in drought, but also floods in summer monsoon periods in Myanmar. There are a lot of evidences for such situations. Authors are encouraged to find more recent literature on El Nino impacts on Myanmar's regional climate as well as the consequences across the country.

12. Page 10. Line 8-10. "During the rainy season….. the cooler winter season'. Actually, there is no cooler winter in Myanmar like Europe. Instead, there is a cold season during November and February. It is better to change "cooler winter season" to "cold season".

13. Page10.Line 25. 'Katchin state'. Please correct as 'Kachin state'

14. Page10.Line 21. Use 'Myanmar' instead of "Burma'.

15. Page 11. Line 5. Please make sure the unit of average annual discharge. Is it 420 km3 per year? Even though we say "average daily discharge", the unit would be m3/sec, not m3/day. Therefore, attention should be paid to the units.

16. Page 11. Line 8. The expression "the suspended load to be 325+/- 57 x106t" needs a description. What is "t" in this expression? ton? 106 is wrong. Correct it.

17. Page 11. Line 9. In the phrase "ca. 1500km from Yangon", is "ca.' an English expression or a German word? It is better to use "approximately 1500 km".

18. Page 14. Line 13-14. I don't get a clear view on the clause "For the future, …..projected to increase." A clear meaning should be provided: The occurrence of 100-year floods is more likely to happen in future (or) the magnitude of 100-year floods likely to increase (or) other meanings? Please revise the sentence.

19. Page 14. Line 24. What is "14 mio people"?  Is it "14 million people"? Please correct it.

20. Section 4 Flora and fauna. Key biodiversity area map should be given. As far as I know such kind of map has already been prepared. Then you can discuss how rich the biodiversity is in Myanmar and how vulnerable they are to climate change, which could also be one of major threats.

21. Section 5.1 Agricultural land use. The authors cited mostly FAO literature regarding Myanmar's agriculture. Annually, the Ministry of Agriculture and Irrigation in Myanmar issues agricultural statistic which is available only in hard copy. Nonetheless, it is recommended that such a reference should be cited. Land use is an important factor worth to be known for various aspects, such as agriculture, flood protection, climate

change adaptation, socio-economic development etc. The authors can refer to the previous studies emphasizing Myanmar's land use. If there is difficulty in finding relevant literature, the authors can use global land use data which is freely available, and do necessary analysis for providing land use information.

22. Page 19. Line 10. Citation "(data from 2000)" is not accurate. Try to provide a legal citation. Otherwise. Please omit it.

23. Section 5.2.1 Central Dry Zone. In this section, only pump irrigation is highlighted for farming in dry zone. Of course, there are pumping systems dedicated to farming in this area. However, it doesn't play a role compared to storage reservoirs from which irrigation water is delivered via canal networks to the irrigated fields. For farming in this central dry zone, there are several storage reservoir projects implemented by the Irrigation Department. The authors are encouraged to focus more on such kind of irrigation system rather than pumping. Then, the discussion could focus on how important these storage reservoirs are to farming in dry areas when drought comes.

---

## Author Comment (AC1) · 30 Mar 2016

Dear reviewer,

first of all, thank you for your critical review. We fully agree with most of your notes, particularly concerning the scientific requirements of a paper in HESS. Our aim was to summarize the water-related relevant information on environmental and social aspects within the Ayeyarwady Basin in Myanmar, in order to use these information as a basis for scientific research. However, we agree that the structure along the socio-hydrologic elements is doubtful in this context, particularly because we didn't draw our own conclusions at the end. Poor data records and the lack of internationally accessible publications are a major challenge for this review. However, you are right to criticize the use of some individual references. Altogether, we would be grateful to get a second

chance in submitting a completely revised version of our paper (incl. updated figures and literature review).

Best Linda Taft
* * *

---

## Author Response (AR1)

Responses to the comments on „ A review of current and possible future human-water interactions in Myanmar's river basins" by L. Taft and M. Evers

**Responses to Anonymous Referee #1**

In spite of my enthusiasm for the paper, I am not sure that the current manuscript can stand alone as a research article. At best it is a literature review or weakly formulated opinion article.  I would like to see the idea developed some more, before it can be published.

In our revised version of the article we have developed the idea towards the identification of key indicators for human-water interactions and changes in Myanmar's river basins.

I understand that the idea is at a concept stage, not even a proposal stage, and there are no research results to present. Still I would like the authors to take the idea further, into at least a proposal stage.  In two recent socio-hydrology papers, we have seen the recommendation to do "framing" of a socio-hydrologic problem.  Framing involves using available information to identify a phenomenon and the domain/scale,  putting together a perceptual model, choosing state variables, causal factors that affect the state variables, developing functional relationships, estimating parameters, and finally model parameter estimation. At the very least the authors could start with starting with a narrative and progressing to the stage of developing a perceptual model (and some hypotheses for identified phenomena), then it will qualify to be a publishable research article.  It is disappointing that the paper has only two figures, one borrowed from

Sivapalan et al.  (2014) and the other a map of Myanmar.  There needs to be more substance.

Therefore, this calls for major revision of the paper.

For more ideas on "framing" I encourage the authors to look up two recent papers (and references found therein).

The socio-hydrology approach which we tried to apply in the first paper version has been omitted because at the current stage we do not have enough reliable data on socio-hydrological research in Myanmar. For this reason we follow another approach in the revised version: the eDPSIR concept, since this framework is more suitable to identify environmental key indicators based on the reviewed literature. We have therefore followed a completely another approach in the revised version.

**Answers to Referee #2**

The paper describes human and water systems in Myanmar. The topic is relevant and, given the focus on a very dynamic system, I think that this paper has a great potential.

Yet, while the title suggests a focus on human-water interactions, e.g. socio-hydrology, this review is mainly a description of either natural or social processes. Very little is said about the reciprocal effects, mutual interactions and feedback loops between nature and society. I would not recommend its publication on HESS because I see little value of the current manuscript from a scientific viewpoint. For instance, the authors often quote Sivapalan et al. (2014) and they also report their conceptual framework (Figure 1). In the paper, however, there is no real attempt to follow it and show it with reference to Myanmar. Section 2, 3 and 5 do follow the three elements of the feedback loop (with in between a "out-of-the-blue" section 4 about flora and fauna), but they do not describe how each element impacts or responds to changes of the other elements, i.e. human-water interactions. In other words, the arrows of Figure 1

(which are, in my opinion, the essence of the referred paper of Sivapalan et al., 2014)

are not sufficiently reflected into the text. Addressing this point (i.e. describing the interplay of the three elements, rather than just the three elements) can be a way out to make this paper scientifically interesting and potentially publishable. More minor points:

1) Reference to recent literature about human-water interactions is also very limited, see e.g. 2015 WRR debate, tons of HESS papers on the subject and recent socio- hydrological studies about Bangladesh, a neighbouring country, China and Australia.

2) English should be improved, as there are numerous typos. 3) I have some more technical comments that might follow if major concern is addressed.

We have completely revised the paper (structure, content, figures) and we do not follow the socio-hydrological
concept any more. English has been improved and literature on human-water interactions has been added.

**Comments on Referee #3**

This manuscript addresses an important issue on human-water interactions in Myanmar's river basins. Authors'
efforts are highly appreciated. However, the manuscript is found to be merely a review which is too much
dependent only on the literature cited. Some of the references are not updated ones, and the authors did not
seem to make a new analysis at least either based on available information or on their own survey. There are a
lot of shortcomings with respect to the review of current water resources sectors and related ones. The
manuscript needs a thorough revision as a whole, and some validations, if possible based on own survey, are
also necessary.

Of course, I agree with that there are very few research publications about every sector of Myanmar, and it is
also very difficult to get access to the published documents many of which are only available within the country. Consequently, the discussion could result a departure from the existing conditions in the country.
Moreover, scarcity on reference information is not an excuse for a research which could be published to world-
wide readers.

I do not recommend the manuscript is suitable for further publication in its present form.

Specific comments:

1. In Abstract, what does the "sound knowledge" mean for? It must be clarified.

We have changes this term into "detailed" knowledge and explained that we mean the understanding and
knowledge which is essential for doing research.

2. In abstract, line 9-12. The meaning of the clause is not clear. "Though" should be omitted in the sentence.

Done.

3. In Abstract, line 17-19. "On the one hand" must be changed to "On one hand".

We don't see why, because both variations are common in the English language.

4. In the introduction, line 16. "Burmese" must be changed to "Myanmar". Like this other places must be
changed.

"Burmese" is the correct and official term for people living in Myanmar. We decided not to change it.

5. Page 4,line 21. Source citation should not be used such as BBC News.

We deleted this reference and refer now to a high-ranked journal (an article in The Lancet).

6. Page 5, line 21-23. The clause "We hypothesize…. Permanently change" is not clear. What are the all
components? Are they interacting with each other? Which driving forces do authors mean? Please provide
examples (such as regional climate change?).

We have given some examples at this text passage.

7. Page 6, line 19. Myanmar is not the largest country in Southeast Asia with respect to overall country's area,
and it is the second largest country after Indonesia which comprises of many large islands. However,
considering the size of successive land, Myanmar owns the largest land size in Southeast Asia. Therefore, the
authors must provide the clear information to readers. Since the sentence (Page 6, line: 18-21) has no citation,
and therefore it must be authors' finding or knowing via some calculations or using existing maps. Please make
sure what the real message is for the readers.

We have deleted this sentence because there is no real message for the readers.

8. Fig. 2. The quality of the figure is not good. The legends for administration boundary and state boundary are
not clear. Different colors for these legends should be used. Moreover, only three major cities are given on the
map, pointing with red circles, but the legends are not provided. If it is not necessary, they can be omitted. If
not, please provide necessary legends for the major cities mentioned on the map. Overall, the quality of the
figure can be improved by GIS techniques, providing clear information, labels and associated legends.

We have created a completely new overview map.

9. Section 2.1. Page 6. How do you decide the important physical features? Only two features are given, i.e.
physiographic characteristics and soil types which are retold from previous studies. How about others, such as
geology?

We do have referred to some geological features which are, from our point of view, essential, i.e. the
formation of the mountain ranges, the plateaus. There is of course a relation between geology, soils, climate and land use. However, we do not see that more knowledge on geological formations is relevant at this very
beginning of our studies.

10. Page 8, line 4-13. Five sub-climate regions for Myanmar are mentioned according to Koeppen-Geiger
classification. It is better to provide a figure showing the country's map with these five sub-climate boundaries.

Because it is just a side note for interested readers, we decided not to provide an extra map. Further
information can be found in the cited reference.

11. Page 9. Line 20-22. The citation is too general. El Nino events not only results in drought, but also floods in
summer monsoon periods in Myanmar. There are a lot of evidences for such situations. Authors are
encouraged to find more recent literature on El Nino impacts on Myanmar's regional climate as well as the
consequences across the country.

We agree that the citation might be too general. However, we would like to know on which research the above
mentioned statement is based. El Nino periods typically cause drier conditions in SE-Asia. Whats more, the
Indian Monsoon circulation generally weakens leading to increased dryness in South Asia. What is the effect
behind flood events caused by El Nino? Possibly, El Nino results in more intense cyclones which could affect the
delta and the coast line.

12. Page 10. Line 8-10. "During the rainy season….. the cooler winter season'. Actually, there is no cooler winter
in Myanmar like Europe. Instead, there is a cold season during November and February. It is better to change
"cooler winter season" to "cold season".

Agreed and changed.

13. Page10.Line 25. 'Katchin state'. Please correct as 'Kachin state'

Done.

14. Page10.Line 21. Use 'Myanmar' instead of "Burma'.

Done.

15. Page 11. Line 5. Please make sure the unit of average annual discharge. Is it 420 km3 per year? Even though
we say "average daily discharge", the unit would be m3/sec, not m3/day. Therefore, attention should be paid
to the units.

Units and reference changed.

16. Page 11. Line 8. The expression "the suspended load to be 325+/- 57 x106t" needs a description. What is "t"
in this expression? ton? 106 is wrong. Correct it.

It is 106 t, we have corrected it. t=tons, and our opinion is, that 't' for 'tons' in this context has not to be
explained, it should be intelligible for all.

17. Page 11. Line 9. In the phrase "ca. 1500km from Yangon", is "ca.' an English expression or a German word?
It is better to use "approximately 1500 km".

Agreed and changed.

18. Page 14. Line 13-14. I don't get a clear view on the clause "For the future, …..projected to increase." A clear
meaning should be provided: The occurrence of 100-year floods is more likely to happen in future (or) the
magnitude of 100-year floods likely to increase (or) other meanings? Please revise the sentence.

Done.

19. Page 14. Line 24. What is "14 mio people"? Is it "14 million people"? Please correct it.

Done.

20. Section 4 Flora and fauna. Key biodiversity area map should be given. As far as I know such kind of map has
already been prepared. Then you can discuss how rich the biodiversity is in Myanmar and how vulnerable they
are to climate change, which could also be one of major threats.

Since biodiversity is not the focus of our research, we decided not to redraw such a map. Of course,
biodiversity is affected by human-water interactions and could be one of the environmental indicators.

21. Section 5.1 Agricultural land use. The authors cited mostly FAO literature regarding Myanmar's agriculture.
Annually, the Ministry of Agriculture and Irrigation in Myanmar issues agricultural statistic which is available
only in hard copy. Nonetheless, it is recommended that such a reference should be cited. Land use is an
important factor worth to be known for various aspects, such as agriculture, flood protection, climate.

We do not have these statistics from the Ministry of Agriculture, neither in hard copy or digitally. Hence, how
can we cite this reference since we do not know the contents? Land use is indeed an important factor and it is
of course part of our current and future research. At the moment we try to analyze remote sensing data in
order to get our own land use information and we furthermore, try to get the statistics from the ministries.

[revised manuscript text omitted]

---

## Author Response (AR2)

Report #1

The paper paper has been significantly improved during the first round of review. As I stated on the first version, the topic is relevant and, given the focus on a very dynamic, but poorly explored system, I think that this paper has a great potential. However, I must say that my points were not completely addressed. For instance, while the title suggests a focus on human-water interactions, this review is mainly a description of either natural or social processes. Very little is said about the reciprocal effects, mutual interactions and feedback loops between nature and society. The reference to social-ecological system and eDPSIR instead of socio-hydrology does not change this limitation. Examples of feedbacks are only mentioned in the introduction as general dynamics, but there is no a single concrete example of human-water system dynamics emerging from the feedback mechanisms between social and hydrological processes in Myanmar. I guess it is possible to find them, so I would like to invite again the authors to provide at least a qualitative description of them, e.g. narrative. I am afraid that two arrows in Figure 2 cannot justify the publication of this paper.

We fully agree with this major point of criticism. Our initial aim was to provide a review based on existing reports and literature and not fully based on own research results (because we are still at the beginning of our research). When one is starting a research project, one of the first steps can be to review the current status of the related topic, thus that has been our initial aim, to collect all the relevant information in order to do human-water research in Myanmar. But we see this critical point, absolutely. Thus, we made a try to give a concrete example related to alluvial farming (farming in the floodplains and on sandbars along the Ayeyarwady River in the dry zone). We think that this example is very suitable to demonstrate concrete human-water interactions and that this example is worth to investigate more in detail in the near future. Myanmar is a country which shows quite dynamic human-water processes and changes at the moment, and due to the fact that qualitative and quantitative research results are lacking, our text provides an overview on natural and social basic information what is strongly needed before starting concrete research projects. However, these information have not been published in summary anywhere in an international review paper to our best knowledge. We feel confident that the concrete example which we have added now, provides a good insight into human-water aspects in Myanmar.

More minor points: 1) Reference to recent literature about human-water interactions is still very limited. I think this should be done regardless the reference to socio-hydrology or eDPSIR framework.

There is a large number of existing literature about human-water interactions. But we didn't want to review all these publications which (at least the majority) point to the fact that both natural sciences and social sciences (interdisciplinarity, transdisciplinarity) are required to investigate human-water related aspects. We think that these aspects have been reviewed in detail enough elsewhere. We wanted to focus on the question, what is going on in Myanmar, what are the system relevant factors when we start doing research in Myanmar, what are the natural and social basic conditions in the country, what can be the starting point of human-water research in the country? And our aim was not to provide a research basis for our own research, instead we want to start a scientific network and discourse on human-water research in Myanmar, based on (but of course not only) the compiled state-of-the-art text.

 2) I suggest more consistency in the terminology, e.g. is this about human-water system or human-environment system?

We've checked this and changed some terminology to "human-water" because that is our focus.

3) I have some more technical comments that might follow if major concern is addressed.

[revised manuscript text omitted]